# PPDPF alleviates hepatic steatosis through inhibition of mTOR signaling

Ning Ma [1,10], Yi-Kang Wang[1,10], Sheng Xu[1], Qian-Zhi Ni[1], Qian-Wen Zheng[1,2], Bing Zhu[1], Hui-Jun Cao[1], Hao Jiang[1], Feng-Kun Zhang[1], Yan-Mei Yuan[1], Er-Bin Zhang[1], Tian-Wei Chen[1], Ji Xia[1], Xu-Fen Ding[1], Zhen-Hua Chen[3], Xiu-Ping Zhang[4], Kang Wang[3], Shu-Qun Cheng[3], Lin Qiu[1], Zhi-Gang Li[5], Yong-Chun Yu[6], Xiao-Fan Wang[7], Bin Zhou[8], Jing-Jing Li [1✉] & Dong Xie [1,2,9✉]

Non-alcoholic fatty liver disease (NAFLD) has become the most prevalent chronic liver disease in the world, however, no drug treatment has been approved for this disease. Thus, it is urgent to find effective therapeutic targets for clinical intervention. In this study, we find that liver-specific knockout of PPDPF (PPDPF-LKO) leads to spontaneous fatty liver formation in a mouse model at 32 weeks of age on chow diets, which is enhanced by HFD. Mechanistic study reveals that PPDPF negatively regulates mTORC1-S6K-SREBP1 signaling. PPDPF interferes with the interaction between Raptor and CUL4B-DDB1, an E3 ligase complex, which prevents ubiquitination and activation of Raptor. Accordingly, liver-specific PPDPF overexpression effectively inhibits HFD-induced mTOR signaling activation and hepatic steatosis in mice. These results suggest that PPDPF is a regulator of mTORC1 signaling in lipid metabolism, and may be a potential therapeutic candidate for NAFLD.

[1] CAS Key Laboratory of Nutrition, Metabolism and Food Safety, Shanghai Institute of Nutrition and Health, University of Chinese Academy of Sciences, Chinese Academy of Sciences, Shanghai, China. [2] School of Life Science and Technology, ShanghaiTech University, 393 Middle Huaxia Road, Shanghai, China. [3] Department of Hepatic Surgery VI, Eastern Hepatobiliary Surgery Hospital, Second Military Medical University, Shanghai, China. [4] Department of Hepatobiliary and Pancreatic Surgical Oncology, The First Medical Center of Chinese People's Liberation Army (PLA) General Hospital, Beijing, China. [5] Department of Thoracic Surgery, Section of Esophageal Surgery, Shanghai Chest Hospital, Shanghai Jiao Tong University, Shanghai, China. [6] Shanghai Institute of Thoracic Oncology, Shanghai Chest Hospital, Shanghai Jiao Tong University, Shanghai, China. [7] Department of Pharmacology and Cancer Biology, Duke University Medical Center, Durham, NC, USA. [8] The State Key Laboratory of Cell Biology, CAS Center for Excellence on Molecular Cell Science, Shanghai Institute of Biochemistry and Cell Biology, Chinese Academy of Sciences, University of Chinese Academy of Sciences, Shanghai, China. [9] NHC Key Laboratory of Food Safety Risk Assessment, China National Center for Food Safety Risk Assessment, Beijing, China. [10] These authors contributed equally: Ning Ma, Yi-Kang Wang. ✉email: tide7@163.com; dxie@sibs.ac.cn

Nonalcoholic fatty liver disease (NAFLD) is defined as excessive liver lipid accumulation (triglyceride content more than 5% of liver weight) excluding other competing causes of hepatic steatosis[1–4]. Approximately 25% of the world population has been affected by NAFLD[2,3,5]. NAFLD encompasses a continuum, from simple hepatic steatosis with moderate fatty infiltration to non-alcoholic steatohepatitis (NASH) with focal inflammation. A small portion of NAFLD may progress to advanced fibrosis, cirrhosis, and eventually hepatocellular carcinoma[6,7]. Many drugs have been tested but no medications have yet been approved for NAFLD[8]. The huge burden of NAFLD calls for more comprehensive understanding of the underlying pathological mechanisms, and the identification of new therapeutic targets.

Hepatic de novo lipogenesis (DNL) refers to the liver's synthesis of new fat from other substances, often carbohydrate and specifically fructose. DNL accounts for 25% of total hepatic lipid and plays an important role in the NAFLD development[9–11]. DNL is mainly controlled by lipogenic transcription factor, sterol regulatory element-binding protein1 (SREBP1)[12–14]. Emerging evidence have suggested that the expression of SREBP1 was regulated by mTOR-S6K axis[15–18].

Pancreatic progenitor cell differentiation and proliferation factor (PPDPF) locates on chromosome 20. The function of PPDPF is rarely investigated, only Jiang et al. reported that EXPDF, a zebrafish homolog of PPDPF was a key regulator of exocrine pancreas development[19], Mao et al. reported that PPDPF predicted poor prognosis in HCC[20]. Liu et al. identified PPDPF as a target of circular RNA circ-FOXM1 facilitating cell progression in non-small cell lung cancer[21]. However, the role of PPDPF in hepatic steatosis has not been elucidated.

In this study, we generate a conditional PPDPF knockout mouse strain, and find that PPDPF-LKO mice spontaneously develop fatty liver on chow diets at 8 months. Moreover, PPDPF-LKO enhances HFD-induced hepatic steatosis, and insulin resistance in vivo. Downregulation of PPDPF is also detected in the steatosis liver tissues of patients with NAFLD. We further demonstrate that PPDPF inhibits mTOR signaling pathway via attenuation of Raptor ubiquitination, by interfering with the interaction between Raptor and DDB1, a member of CUL4B–DDB1 complex. Importantly, we identify 51–64 amino acid (aa) residues of PPDPF mediating its interaction with Raptor. Finally, PPDPF overexpression via vein injection of AAV8-PPDPF effectively suppresses HFD-induced mTOR signaling activation and fatty liver. Therefore, our study identifies PPDPF as a negative regulator of lipogenesis by inhibiting mTOR-S6K-SREBP1 signaling and provides a promising therapeutic candidate for NAFLD.

## Results

**Liver-specific PPDPF deficiency leads to development of fatty liver disease and PPDPF expression is downregulated in human NAFLD.** To investigate the function of PPDPF in the liver, we generated liver-specific *PPDPF* knockout mice (Alb-Cre; *PPDPF*^f/f, PPDPF-LKO). The successful knockout of PPDPF was confirmed by qPCR and western blotting using liver samples (Supplementary Fig. 1a). PPDPF-LKO mice are healthy and their livers are not different from wild-type (WT) mice at the beginning (Supplementary Fig. 1b). However, PPDPF-LKO mice had a higher body weight than WT mice at 8 months, and the livers of LKO mice were heavier than those of WT mice (Fig. 1a–c). H&E and Oil Red O staining showed lipid droplets with increased size and serious steatosis in the livers of PPDPF-LKO mice, while the liver of WT mice was healthy, and the lipid droplets were much smaller (Fig. 1d). Accordingly, serum triglycerides (TG) (Supplementary Fig. 1c) and liver TG, NEFA (Fig. 1e, f) were

dramatically increased in PPDPF LKO group. qPCR analysis of liver tissues revealed that mRNA levels of the key molecules in fatty acid synthesis were higher in PPDPF-LKO mice, including *SREBP1, FASN, ACLY, ME,* and *PPARG* (Fig. 1g). Overall, these data suggested that PPDPF-LKO mice were more susceptible to fatty liver disease.

To confirm the clinical impact of PPDPF in NAFLD patients, immunohistochemistry experiments were performed. As shown in Fig. 1h, PPDPF protein expression was markedly lower in steatotic liver tissues from patients with NAFLD when compared with liver tissues from healthy controls.

**Hepatic-specific PPDPF deletion exacerbates HFD-induced hepatic steatosis.** The above data indicated that PPDPF-LKO led to spontaneous development of fatty liver, then we further examined the susceptibility of PPDPF-LKO mice to high-fat diet (HFD)-induced NAFLD. The WT and PPDPF-LKO mice were fed either chow diet or high fat diet (HFD) for 16 weeks. The loss of PPDPF in the liver was verified by western blotting (Supplementary Fig. 2a). The body weight of HFD group was increased when compared with normal chow diet (NC) group (Fig. 2a). Although there was no significant difference in body weight between WT mice and PPDPF-LKO mice upon HFD feeding, the liver weights of PPDPF-LKO mice were increased when compared with WT mice (Fig. 2b), which was due to increased TG and NEFA content in the liver (Fig. 2c, d). Serious steatosis was accompanied with higher TG levels in the serum of PPDPF-LKO mice (Supplementary Fig. 2b). Furthermore, the glucose tolerance test and insulin tolerance test revealed that areas under the curve were increased in PPDPF-LKO mice, indicating that they developed more severe glucose intolerance and insulin resistance than WT mice after 16-week HFD feeding (Fig. 2e). The staining of H&E and Oil Red O disclosed that PPDPF-LKO mice had larger and more lipid droplets in the livers (Fig. 2f). We also observed upregulation of the key genes involved in lipid synthesis, which was consistent with the phenotypes (Fig. 2g).

We also examine the inhibitory effect of PPDPF on lipogenesis at cellular level. We isolated primary hepatocytes from WT and PPDPF-LKO mice, and the knockout efficiency was confirmed by qPCR (Fig. 2h). The primary hepatocytes were treated with palmitic acid (PA) to induce steatosis in vitro. 24 h treatment with PA increased lipid deposition in both control and PPDPF-LKO hepatocytes. However, the increase was much more dramatic in PPDPF-LKO hepatocytes, which was revealed by Oil Red O staining and TG content examination (Fig. 2i, j). We also detected the expression of genes involved in lipogenesis, and found that the mRNA levels of key molecules were much higher in PPDPF-LKO cells (Fig. 2k). In conclusion, hepatocyte-specific PPDPF depletion exaggerated HFD-induced lipid droplet accumulation, glucose intolerance and insulin resistance in vivo, as well as PA-induced steatosis in vitro.

**Reintroduction of PPDPF inhibits the development of NAFLD in LKO mice.** To further confirm the effect of PPDPF on the development of NAFLD, expression of PPDPF was rescued by AAV8-TBG-PPDPF virus via tail vein injection in PPDPF-LKO mice. AAV8-TBG-PPDPF dramatically reduced body weight (Fig. 3a), liver weight (Fig. 3b), hepatic TG (Fig. 3c), NEFA (Fig. 3d), and lipid droplet accumulation (Fig. 3e) at 8 months. Consistently, AAV8-TBG-PPDPF also decreased the expression of lipogenic genes (Fig. 3f).

Next, we explored the effect of reintroduction of PPDPF on NAFLD in the HFD-fed mouse model. We reintroduced PPDPF back into the livers of 10-week-old and HFD-fed PPDPF-LKO mice by injection of AAV8-TBG-PPDPF, followed by HFD

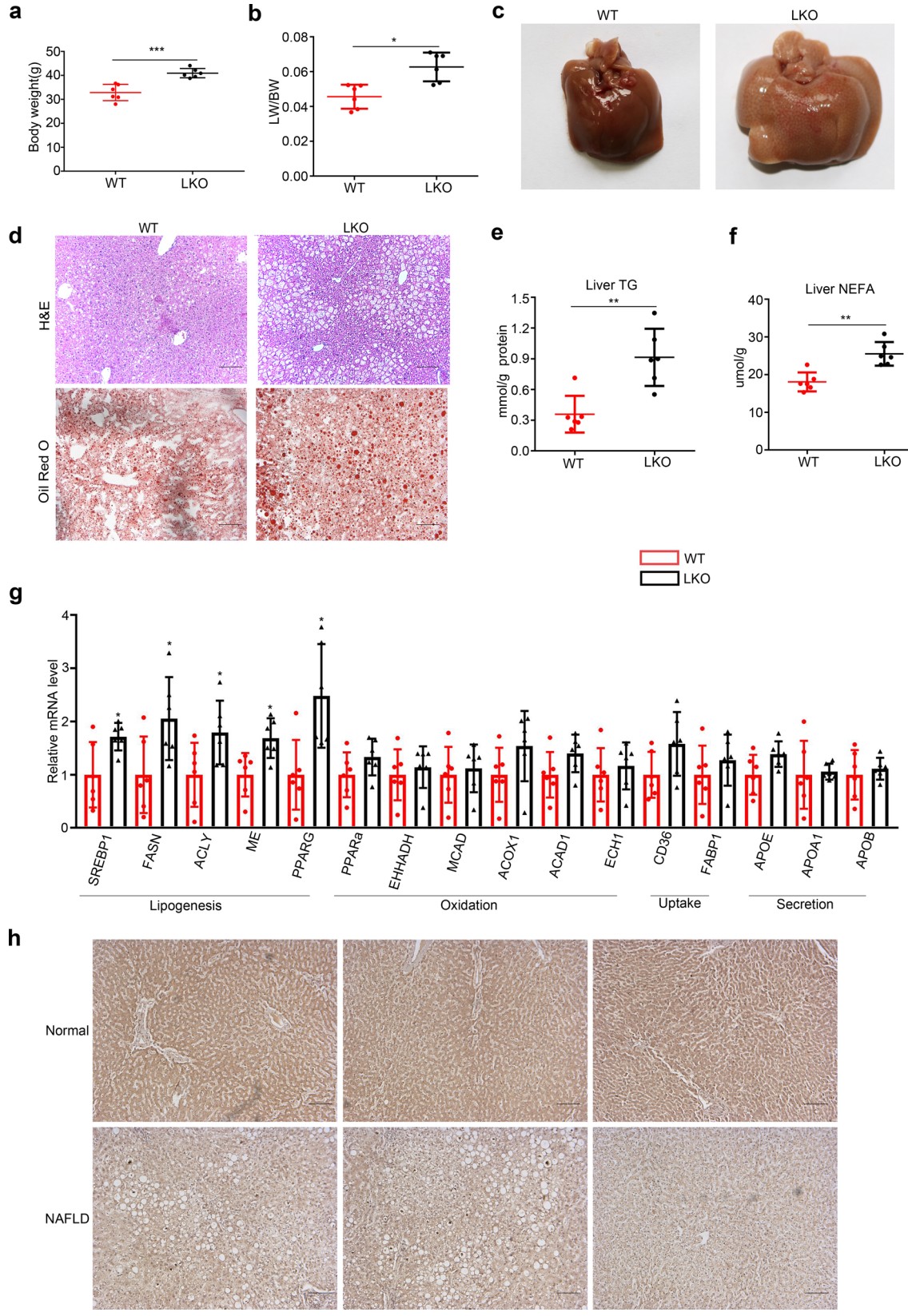

feeding for another 3 months. Compared with AAV8-con, AAV-mediated reintroduction of PPDPF reduced body weight (Supplementary Fig. 3a), liver weight (Supplementary Fig. 3b), liver TG (Supplementary Fig. 3c), liver NEFA (Supplementary Fig. 3d), lipid droplet accumulation (Supplementary Fig. 3e) and the mRNA levels of lipogenic genes (Supplementary Fig. 3f) in PPDPF-LKO mice after 3 months of HFD consumption. In sum, these findings revealed that re-expression of PPDPF hindered HFD-induced NAFLD development in PPDPF-LKO mice, which confirmed the inhibitory effect of PPDPF on NAFLD.

**Fig. 1 Liver- specific PPDPF deficiency leads to the development of fatty liver disease. a** The body weight of PPDPF-LKO ($n = 6$) and WT ($n = 6$) mice at 8 months. Mean ± SEM, \*\*\*$p = 0.0005$ by two-tailed unpaired Student's $t$ test. **b** The ratio of liver-to-body weight (LW/BW) of PPDPF-LKO ($n = 6$) and WT ($n = 6$) mice at 8 months. Mean ± SEM, \*$p = 0.003$ by two-tailed unpaired Student's $t$ test. **c** Liver images of WT and LKO mice at 8 months of age. **d** Images of H&E and Oil Red O staining of liver tissues from the WT and LKO mice at 8 months. Scale bars, 100 μm. **e, f** The triglyceride (TG) and nonestesterified fatty acid (NEFA) in the livers of WT and LKO mice at 8 months of age ($n = 6$ mice for each group). Mean ± SEM, **e** \*\*$p = 0.002$, **f** \*\*$p = 0.001$ by two tailed unpaired Student's $t$ test. **g** The relative mRNA levels of genes involved in lipid metabolism, including lipogenesis genes, oxidation-related genes, lipid uptake, and lipid secretion in WT and LKO mice at 8 months of age ($n = 6$ for each group). The mRNA expression levels of the genes are normalized to that of 18 s. Mean ± SEM, SREBP1: \*$p = 0.025$, FASN: \*$p = 0.035$, ACLY: \*$p = 0.045$, ME: \*$p = 0.012$, PPARG: \*$p = 0.011$ by two-tailed unpaired Student's $t$ test. **h** Representative images of immunohistochemical staining of PPDPF in liver sections from controls and NAFLD patients. Scale bars,100 μm. All experiments were repeated three times independently.

**PPDPF regulates hepatic steatosis via the mTOR signaling pathway**. Next, we explored the mechanisms underlying the inhibition of steatosis by PPDPF. Considering that lipid synthesis was mainly regulated by mTOR-S6K-SREBP1 axis[15,16], we examined the influence of PPDPF on this signaling. As shown in Fig. 4a, the level of phosphorylated p70 S6K was higher in the liver tissues of PPDPF-LKO mice when compared with their counterparts fed a chow diet for 8 months, which was similar in HFD group fed a HFD for 16 weeks (Fig. 4b). In addition, the protein level of cleaved SREBP1, and FASN was also significantly increased in PPDPF-LKO mice, both of which were key molecules in lipid synthesis and downstream effectors of S6K. Furthermore, the level of p-p70 S6K, SREBP1, and FASN increased more dramatically in PPDPF-LKO primary hepatocytes when compared with WT cells (Fig. 4c), while the activation of mTOR-S6K-SREBP1 signaling was impaired by PPDPF overexpression in HepG2 cells upon PA treatment (Fig. 4d). We also found that re-expression of PPDPF in PPDPF-LKO mice decreased the level of p-p70 S6K, SREBP1, and FASN in the livers on either a chow diet (Fig. 4e) or HFD (Fig. 4f).

To verify whether PPDPF affected lipid synthesis through mTOR signaling, isolated hepatocytes from both WT and PPDPF-LKO mice, were treated with mTOR inhibitor Torin1. Oil Red O staining and TG test revealed that lipid deposition was significantly reduced upon Torin1 treatment in PPDPF-LKO cells (Supplememntary Fig. 4a, b). Torin1 treatment led to significantly decreased expression of lipogenic genes in PPDPF-LKO cells at 2 h, while there was no significant change of these genes in WT cells. At 18 h, the decrease in the expression of these genes was more dramatic in PPDPF-LKO cells when compared with WT cells (Supplementary Fig. 4c). These results indicated that PPDPF regulated lipid synthesis through regulation of mTOR signaling.

For further verification, we used another classical mTOR inhibitor Rapamycin in in vivo experiment. 6-month-old WT and LKO mice fed a chow diet were treated with rapamycin, then sacrificed 2 months later. We found that rapamycin treatment significantly reduced body weight (Supplementary Fig. 5a), liver weight (Supplementary Fig. 5b), liver TG (Supplementary Fig. 5c), liver NEFA (Supplementary Fig. 5d) of PPDPF-LKO mice. Consistently, the staining of H&E and Oil Red O showed that rapamycin inhibited the formation of lipid droplets (Supplementary Fig. 5e) and the expression of lipid synthesis-related genes in PPDPF-LKO mice (Supplementary Fig. 5f). As expected, rapamycin also suppressed activation of mTOR signaling pathway in PPDPF-LKO group (Supplementary Fig. 5g). It was worthy to note that Rapamycin showed little effect on WT mice in the above aspects. These results further identified that NAFLD induced by PPDPF loss was caused by activation of mTOR signaling pathway.

**PPDPF directly interacts with Raptor and negatively regulates its ubiquitination**. Since PPDPF is mainly localized in the cytoplasm, we speculated that it may regulate mTOR signaling by interacting with mTOR complex. Therefore, we examined the interaction between PPDPF and principal members of mTOR complex, including mTOR, Raptor, and Rictor. We found that PPDPF was associated with mTOR and Raptor, but not Rictor. Further analysis disclosed that PPDPF could interact with Raptor, and vice versa (Fig. 5a, b). However, mTOR could not be pulled down by PPDPF. Furthermore, interaction between endogenous Raptor and PPDPF was detected in primary hepatocytes (Fig. 5c, d), and GST pull-down assay further confirmed the interaction between PPDPF and Raptor (Fig. 5e). Next, we searched for the critical amino acid residues in PPDPF responsible for its interaction with Raptor. We divided PPDPF into five segments, and fused each segment with GST tag. GST pull-down assay indicated that the 51-64aa of PPDPF mediated its interaction with Raptor (Fig. 5f). To further confirm this finding, we mutated all 51-64aa into Alanine (PPDPF mut). We transfected 293T cells with WT PPDPF and PPDPF mut, respectively, and the immunoprecipitation experiments showed that PPDPF-mut could not interact with Raptor (Fig. 5g, h), which was further confirmed by the GST pull-down assay (Fig. 5i).

mTOR signaling pathway plays essential roles in multiple biological processes, hyperactivation of this pathway is observed in various diseases[22]. In addition to changes at genetic levels, dysregulated posttranslational modifications of key members are also critical[23,24]. Ubiquitination of Raptor was a necessary step in the activation of mTOR signaling pathway[25,26]. Therefore, we examined the effect of PPDPF or PPDPF mut on Raptor ubiquitination in 293T and HepG2 cells. As shown in Fig. 5j and Supplementary Fig. 6a, PPDPF overexpression decreased Raptor ubiquitination, while PPDPF mut had little effect on Raptor ubiquitination. Since it is not clear whether Raptor ubiquitination is involved in response to lipid signal, we treated cells with palmitate. PA stimulation significantly induced Raptor ubiquitination in HepG2 cells. This induction was dramatically inhibited in PPDPF- overexpressing cells, while PPDPF mut showed no inhibitory effect (Supplementary Fig. 6b). Consistently, PPDPF depletion increased PA-induced ubiquitination of Raptor in primary hepatocytes (Supplementary Fig. 6c). Moreover, we also examined the phosphorylation of p70 S6K, the expression of SREBP1 and FASN in HepG2 cells upon PA treatment. Overexpression of PPDPF inhibited activation of mTOR signaling, but PPDPF mut exerted little influence on mTOR signaling (Supplementary Fig. 6d). Therefore, the above data suggested that PPDPF regulated mTOR signaling activation triggered by lipid signal through influencing Raptor ubiquitination.

**PPDPF-Raptor interaction is required for regulation of mTOR signaling and lipid metabolism by PPDPF**. In order to clarify the role of PPDPF-Raptor interaction in the regulation of mTOR signaling and lipid metabolism, AAV8-TBG-PPDPF and AAV8-TBG-PPDPF mut virus were introduced into the livers of PPDPF-LKO mice via tail vein injection. Consistent with the results in

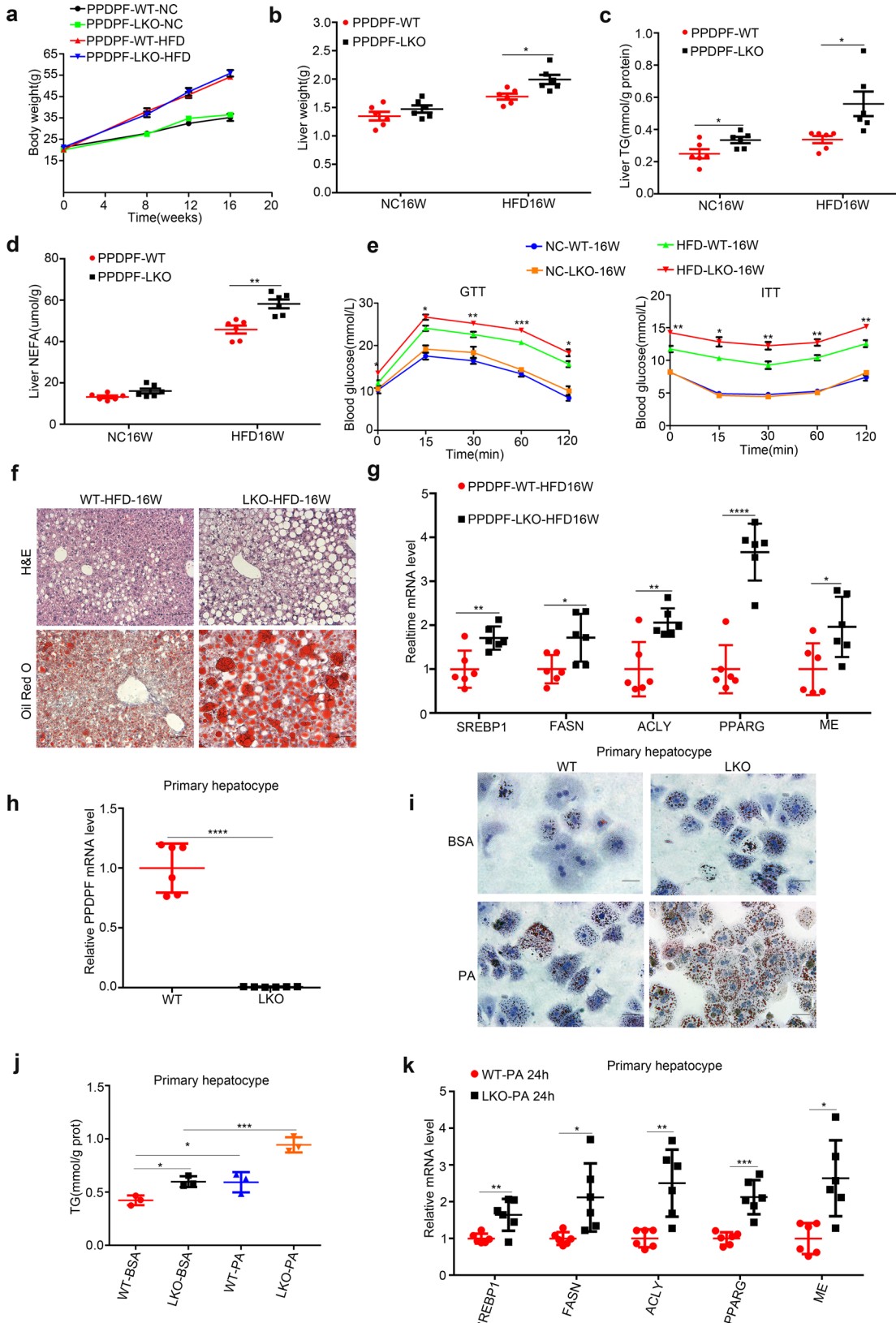

Fig. 3, AAV8-TBG-PPDPF dramatically reduced body weight (Supplementary Fig. 7a), liver weight (Supplementary Fig. 7b), hepatic TG (Fig. 6a), NEFA (Fig. 6b), lipid droplet accumulation (Fig. 6c), and the expression of lipogenic genes (Supplementary Fig. 7c) at 8 months. In contrast, AAV8-TBG-PPDPF mut did not show any rescue effect at 8 months (Fig.6a–c, Supplementary Fig. 7a–c). Moreover, PPDPF overexpression reduced the level of p-p70 S6K, SREBP1, and FASN in PPDPF-LKO mice when compared with controls, while PPDPF mut did not affect mTOR signaling (Fig. 6d).

**Fig. 2 PPDPF deficiency exacerbates HFD-induced hepatic steatosis.** Body weight (**a**) and liver weight (**b**) of WT($n = 6$) and LKO ($n = 6$) mice fed a chow diet or HFD for 16 weeks. Mean ± SEM, *$p = 0.01$ by two tailed unpaired Student's $t$ test. The triglyceride (TG) (**c**) and nonesterified fatty acid (NEFA) (**d**) levels in the livers of WT and LKO mice at the end of 16 weeks of HFD feeding ($n = 6$ mice for each group). Mean ± SEM, (**c**) *$p = 0.032$ (NC16W), *$p = 0.019$ (HFD16W), (**d**) **$p = 0.001$ (HFD16W) by two tailed unpaired Student's $t$ test. **e** GTTs and ITTs are performed in WT and LKO mice after 20 weeks of NC or 21 weeks of HFD feeding ($n = 6$ in each group). Mean ± SEM. See Supplementary Data 1 for statistics. **f** Images of H&E and Oil Red O staining of liver tissues from WT and LKO mice fed HFD for 16 weeks. Scale bars, 100 μm. **g** The mRNA levels of lipogenic genes in the livers of the WT and LKO mice ($n = 6$ in each group). Mean ± SEM, SREBP1: **$p = 0.006$, FASN: *$p = 0.019$, ACLY: **$p = 0.004$, PPARG: ****$p < 0.0001$, ME: *$p = 0.026$ by two tailed unpaired Student's $t$ test. **h** PPDPF expression is measured by qPCR in primary hepatocytes. Mean±SEM, ****$p < 0.0001$ by two-tailed unpaired Student's $t$ test. **i** Oil Red O staining of the primary hepatocytes from WT and LKO mice exposed to 0.4 mM plamitic acid (PA). Scale bars, 100um. **j** Triglyceride quantification of the primary hepatocytes from WT ($n = 3$) and LKO ($n = 3$) mice exposed to 0.4 mM PA. Mean ± SEM, *$p = 0.01$ (WT-BSA Vs LKO-BSA), *$p = 0.04$ (WT-BSA Vs WT-PA), ***$p = 0.002$ (LKO-BSA Vs LKO-PA) by two tailed unpaired Student's $t$ test. **k** The relative mRNA levels of the indicated molecules in cells from WT ($n = 6$) and LKO ($n = 6$) mice. Mean ± SEM, SREBP1: **$p < 0.005$, FASN: *$p = 0.015$, ACLY: **$p = 0.003$, PPARG: ***$p = 0.0002$, ME: **$p = 0.0049$ by two-tailed unpaired Student's $t$ test. All experiments were repeated three times independently.

We also conducted the similar experiments in HFD-fed mouse model. We reintroduced PPDPF and PPDPF mut back into the livers of 10-week-old HFD-fed PPDPF-LKO mice, followed by HFD feeding for another 3 months. Compared with AAV8-con, AAV8-mediated reintroduction of PPDPF reduced body weight (Supplementary. 7d), liver weight (Supplementary Fig. 7e), liver TG (Fig. 6e), liver NEFA (Fig. 6f), lipid droplet accumulation (Fig. 6g), and the mRNA level of lipogenic genes (Supplementary Fig. 7f) of PPDPF-LKO mice after 3 months of HFD consumption, while AAV8-TBG-PPDPF mut could not alleviate HFD-induced steatosis (Fig. 6e–g, Supplementary Fig. 7d–f). Consistently, PPDPF overexpression reduced the level of p-p70 S6K, SREBP1 and FASN in mice when compared with controls, but PPDPF mut did not (Fig. 6h). In sum, these findings revealed that reintroduction of PPDPF hindered HFD-induced NAFLD development in PPDPF-LKO mice, while PPDPF mut lost the rescue effect, indicating that the interaction between PPDPF and Raptor was important for the regulation of mTOR signaling and lipid metabolism by PPDPF.

**PPDPF disrupts DDB1-Raptor interaction and blocks Raptor ubiquitination.** In order to figure out how PPDPF regulates the ubiquitination of Raptor, mass spectrometry analysis was performed using 293T cells expressing 3xFlag-PPDPF (Supplementary Fig. 8a). We found that many of the candidate interaction partners of PPDPF were E3 ligases, i.e., HUWE1, TRIM21, CUL4B and so on. Since CUL4B-DDB1 is the known E3 ligase for Raptor[27], and DDB1 is responsible for coupling the target protein to E3 ligase (Supplementary Fig. 8b), we performed Co-immunoprecipitation (IP) assay to examine the interaction between DDB1 and PPDPF. As shown in Fig. 7a and b, interaction between PPDPF and DDB1 was detected by either immunoprecipitation of Flag-DDB1 or HA-PPDPF. Moreover, GST pull-down further confirmed the interaction between PPDPF and DDB1 (Fig. 7c). In Fig. 7d, DDB1 overexpression enhanced the ubiquitination of Raptor, while PPDPF alleviated this effect, indicating that PPDPF really regulated the ubiquitination of Raptor by CUL4B-DDB1.

Since the activation of mTORC1 is dependent on ubiquitination of Raptor by CUL4B-DDB1, we hypothesized that PPDPF may interfere with Raptor-DDB1 interaction. To identify this possibility, we overexpressed PPDPF in 293T cells, and then performed Co-immunoprecipitation (IP) assay with Raptor antibody. As shown in Fig. 7e, interaction between Raptor and DDB1 was weakened by PPDPF. Meanwhile, the interaction between Raptor and mTOR was also disturbed by PPDPF, which may be due to the weakened Raptor ubiquitination by PPDPF. The similar result was obtained in HepG2 cells (Fig. 7f). We also examined whether PPDPF mu could disrupt the DDB1-Raptor interaction in both 293T and HepG2 cells. As expected, the

Raptor interaction-deficient PPDPF mutant could not interfere with DDB1-Raptor interaction (Supplementary Fig. 9a and b). Furthermore, we performed PA treatment in control and PPDPF-overexpressing HepG2 cells, and examined the interactions between Raptor and DDB1, mTOR. As shown in Fig. 7g, PA treatment rapidly enhanced Raptor-mTOR interaction, while it had less effect on Raptor-DDB1 interaction. PPDPF overexpression dramatically suppressed Raptor-mTOR and Raptor-DDB1 interaction, either at resting state or during lipid stimulation, suggesting that the interference of Raptor-DDB1 interaction by PPDPF may affect mTOR signaling activation by lipid signal. For further verification, we isolated primary hepatocytes from PPDPF-WT and PPDPF-LKO mice, and Co-IP assay revealed that PPDPF knockout markedly enhanced the interaction between Raptor and DDB1, mTOR (Fig. 7h). Furthermore, PPDPF suppressed the interaction between Raptor and DDB1 or mTOR in a dose-dependent manner (Supplementary Fig. 8c). Moreover, rescue of PPDPF expression suppressed the interaction between Raptor and DDB1/mTOR in vivo (Fig. 7i), which further verified the mechanism disclosed in vitro. Collectively, our data demonstrated that Raptor ubiquitination was enhanced by lipid signal. PPDPF inhibited Raptor ubiquitination by disrupting the interaction between Raptor and DDB1, which may affect Raptor-mTOR interaction and suppress mTOR signaling activation by lipid stimulation.

**PPDPF overexpression reduces HFD-induced hepatic steatosis.** The inhibitory effect of PPDPF on lipogenic signaling downstream of mTOR prompted us to evaluate its therapeutic potential in NAFLD. WT mice were injected with AAV8-TBG-PPDPF or AAV8-TBG control virus via the tail vein at 6 weeks old. Both PPDPF-overexpressing mice and control mice were fed a chow diet (NC) and HFD diet, respectively.

12 weeks later, there was no significant difference in the physiological parameters between PPDPF-overexpressing mice and control mice in NC group. However, in HFD group, the body weight, liver weight, and hepatic lipid content were decreased in PPDPF-overexpressing mice when compared with control mice (Fig. 8a–d). Consistently, the serum TG was decreased in AAV8-TBG-PPDPF mice fed HFD (Supplementary Fig. 10a). Also, H&E and Oil Red O staining revealed that the accumulation of lipid droplets was decreased significantly in PPDPF-overexpressing mice (Fig. 8e). In addition, PPDPF overexpression effectively improved HFD-induced glucose intolerance and insulin resistance (Fig. 8f, g). Furthermore, the mRNA levels of lipogenesis-related genes were significantly decreased in AAV8-TBG-PPDPF mice (Fig. 8h). Consistently, PPDPF overexpression reduced the level of p-p70 S6K, SREBP1, and FASN in mice fed HFD when compared with controls (Fig. 8i).

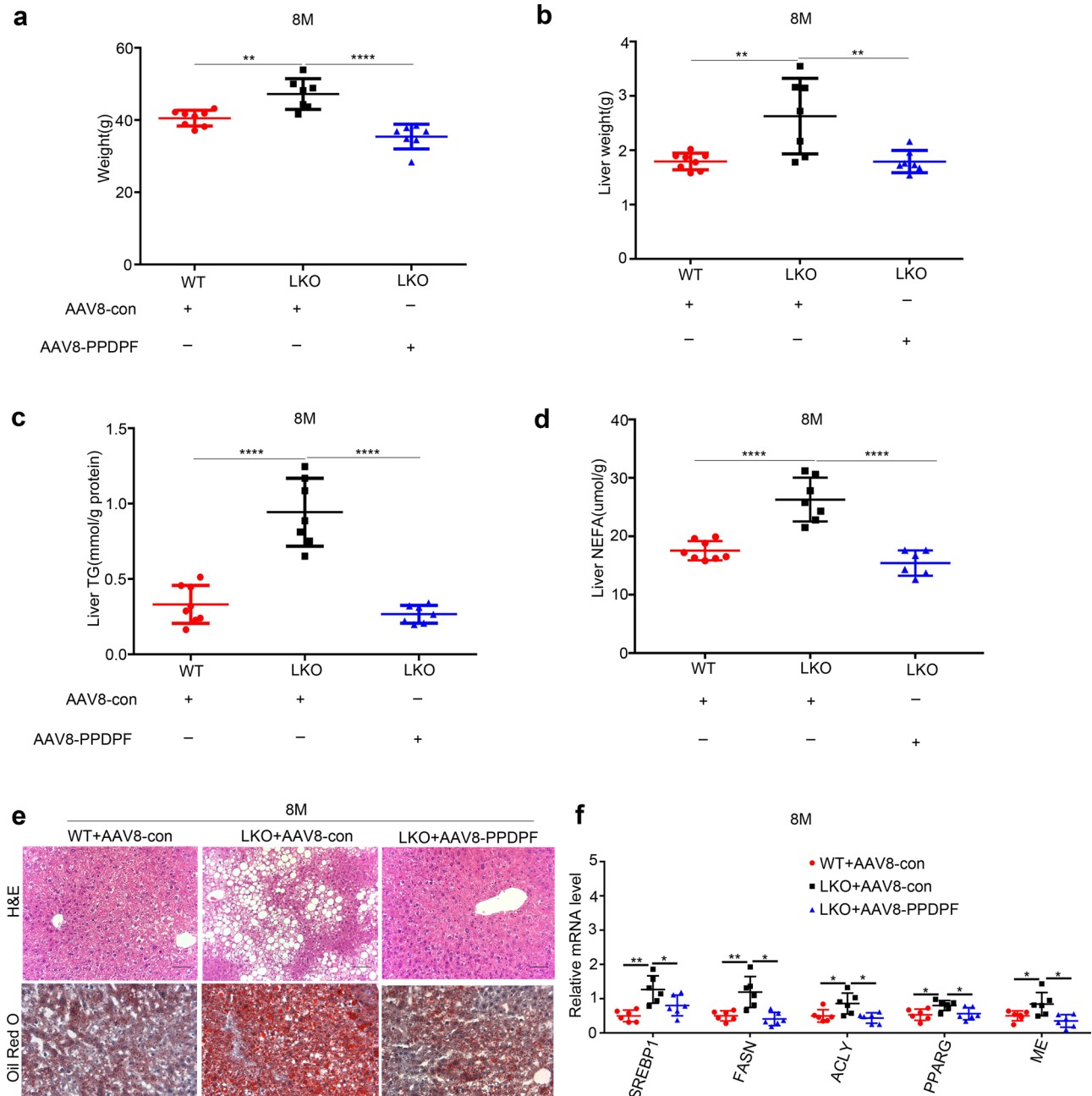

**Fig. 3 AAV8-mediated reintroduction of PPDPF rescues the phenotype of PPDPF-null in PPDPF-LKO mice fed a chow diet for 8 months.** The body weight (**a**), liver weight (**b**), TG test (**c**) and NEFA test (**d**) of WT + AAV8-con (n = 8), LKO + AAV8-con (n = 7), and LKO + AAV8-PPDPF (n = 7) mice at 8 months on chow diets. Mean ± SEM, (**a**) **p = 0.0017 (WT + AAV8-con Vs LKO + AAV8-con), ****p < 0.0001 (LKO + AAV8-con Vs LKO + AAV8-PPDPF), (**b**) **p = 0.0055 (WT + AAV8-con Vs LKO + AAV8-con), **p = 0.01 (LKO + AAV8-con Vs LKO + AAV8-PPDPF), (**c**) ****p < 0.0001 (WT + AAV8-con Vs LKO + AAV8-con), ****p < 0.0001 (LKO + AAV8-con Vs LKO + AAV8-PPDPF), (**d**) ****p < 0.0001 (WT + AAV8-con Vs LKO + AAV8-con), ****p < 0.0001 (LKO + AAV8-con Vs LKO + AAV8-PPDPF) by two-tailed unpaired Student's t test. **e** Representative images of H&E and Oil Red O staining of liver sections from the mice injected with indicated adenovirus at 8 months. Scale bars, 100 um. **f** The mRNA expression levels of lipogenesis-related genes in WT + AAV8-con (n = 6), LKO + AAV8-con (n = 6) and LKO + AAV8-PPDPF (n = 6) mice at 8months. Mean ± SEM, SREBP1: **p = 0.0015, *p = 0.0463; FASN: **p = 0.0050, *p = 0.0029; ACLY: *p = 0.0322, *p = 0.0123; PPARG: *p = 0.0134, *p = 0.0281; ME: *p = 0.459, *p = 0.011 by two-tailed unpaired Student's t test. All experiments were repeated three times independently.

To explore the effect of PPDPF in human hepatocytes, we established a HepG2 cell line stably overexpressing PPDPF (Supplementary Fig. 10b). As expected, PPDPF decreased PA-induced lipid droplet accumulation, which was disclosed by Nile Red staining and TG content examination (Supplementary Fig. 10c, d). The expression of genes involved in lipogenesis was significantly lower in PPDPF-overexpressing HepG2 cells when compared with control cells upon PA treatment (Supplementary Fig. 10e). Therefore, the phenotype in PA-induced hepatic steatosis model in vitro further confirmed the in vivo observation. These data suggested the protective effect of PPDPF against HFD-induced hepatic steatosis, and indicated its therapeutic potential in NAFLD.

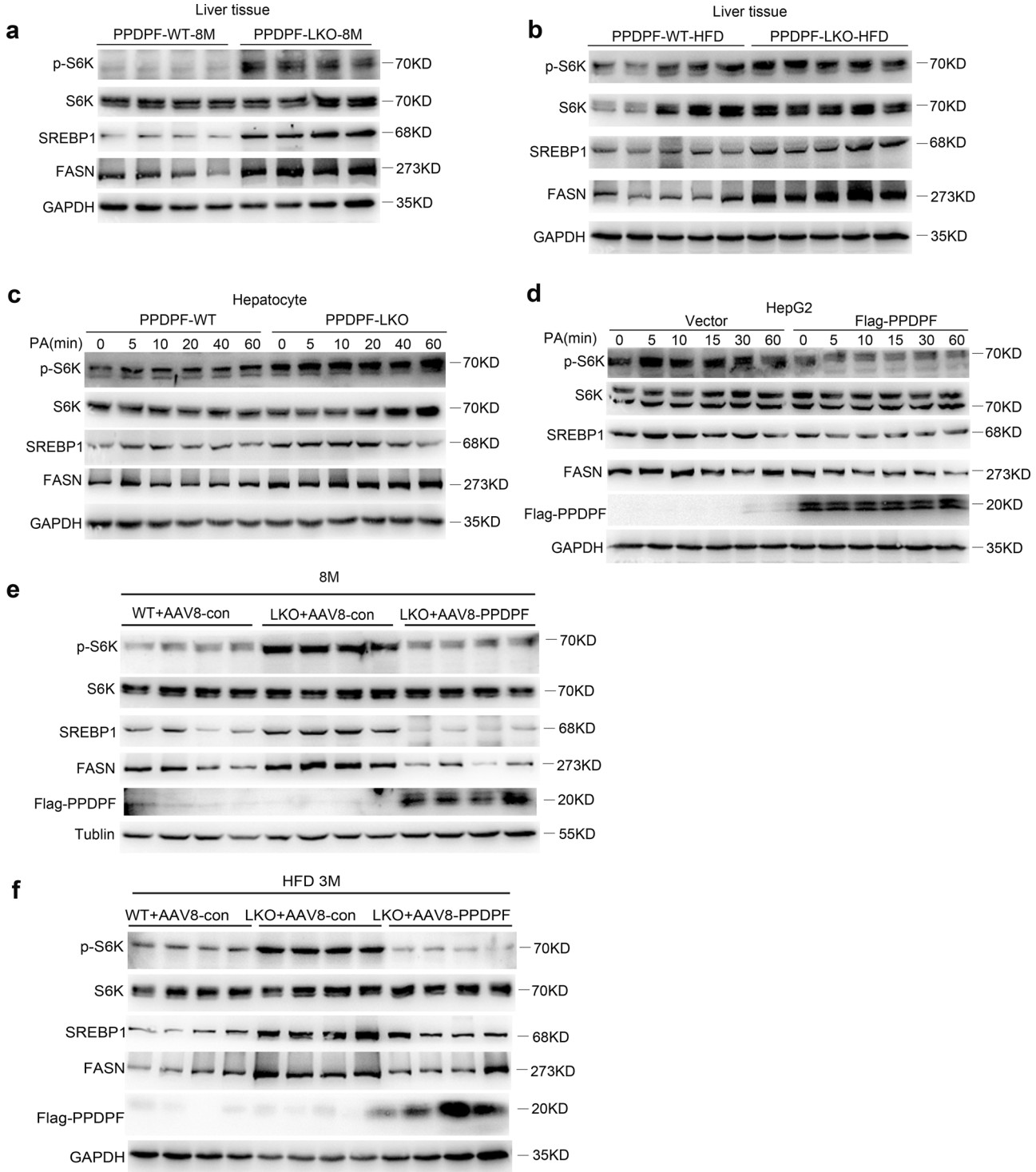

**Fig. 4 PPDPF represses mTOR signaling pathway during the development of hepatic steatosis.** Western blot analysis of phosphorylated S6K, total S6K, cleaved-SREBP1, and FASN in the livers from NC (**a**) ($n = 4$ per group) or HFD (**b**) ($n = 5$ per group) fed mice in PPDPF-WT and PPDPF-LKO groups. Western blot analysis of phosphorylated S6K, total S6K, cleaved-SREBP1, and FASN in primary hepatocytes (**c**) and HepG2 (**d**) cells upon PA treatment. **e** Expression of p-S6K, S6K, SREBP1 and FASN in liver samples of WT + AAV8-con, LKO + AAV8-con and LKO + AAV8-PPDPF mice in rescue experiment ($n = 4$ per group). **f** Expression of p-S6K, S6K, SREBP1, and FASN in liver samples of WT + AAV8-con, LKO + AAV8-con and LKO + AAV8-PPDPF mice in rescue experiment after 3-month HFD feeding ($n = 4$ per group). All experiments were repeated three times independently.

## Discussion

NAFLD has become the most common chronic liver disease in the western world, accounting for about 25% of the total population, and will continue to increase in the future[1,5,6,8,28,29]. NAFLD can progress through more severe NASH, cirrhosis, and, lastly, HCC[30–33]. It is now considered as a multifactorial condition that leads not only to increased liver-related mortality but also to increased risk of the development of type 2 diabetes mellitus and cardiovascular disease[34]. So far, there is no US Food and Drug Administration (FDA)-approved treatment for

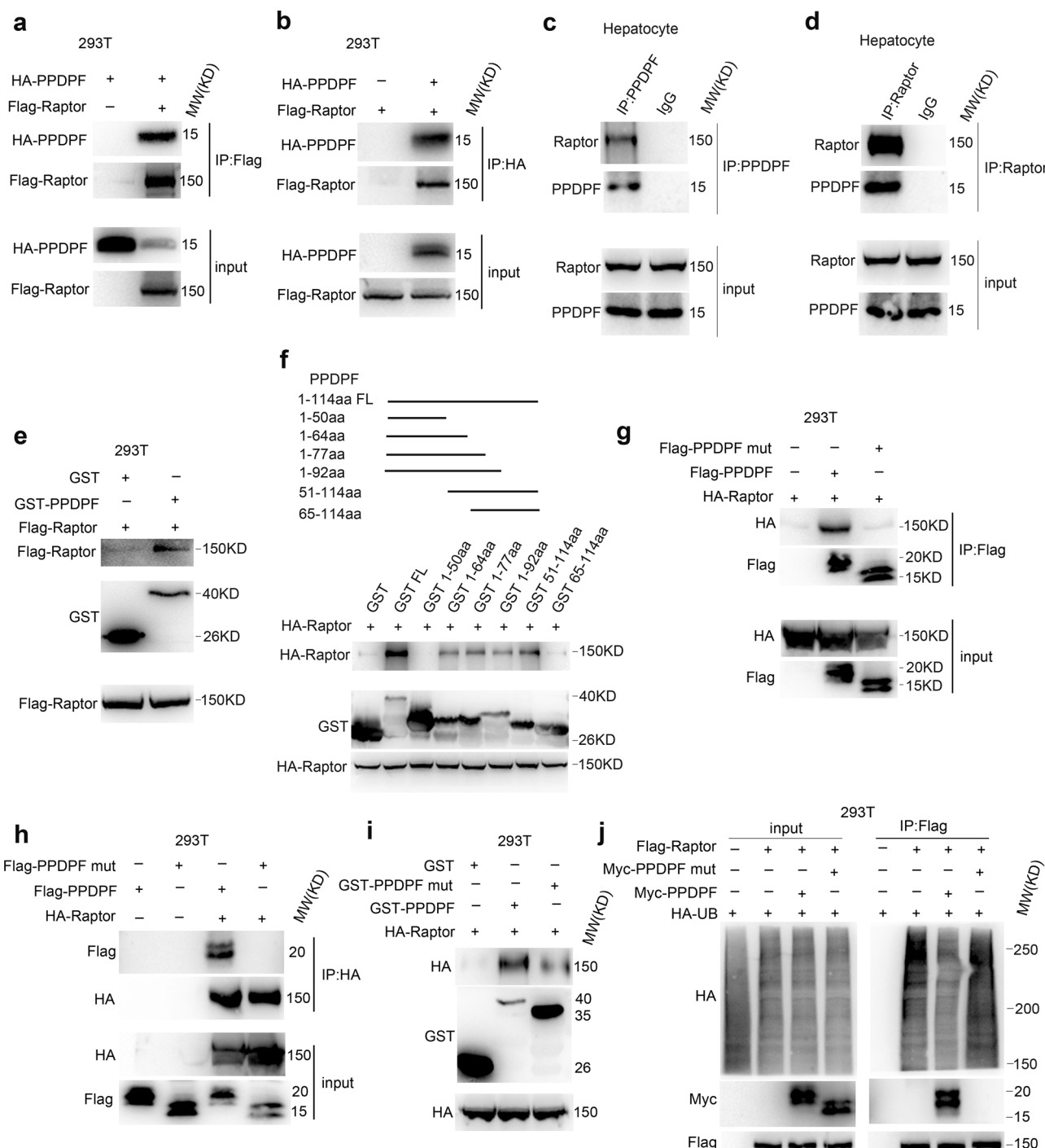

**Fig. 5 PPDPF interacts with Raptor and inhibits the ubiquitination of Raptor. a, b** Co-immunoprecipitation assay in 293T cells co-transfected with Flag-Raptor and HA-PPDPF. **c, d** Examination of the interaction between endogenous PPDPF and Raptor by co-immunoprecipitation. **e** PPDPF interacts with Raptor in vitro, purified GST was used as a control. **f** GST-pull down assay using different truncation mutants of PPDPF. **g, h** 293 T cells were transfected with control vector and PPDPF WT, PPDPF mut, HA-Raptor plasmid, respectively. After 48 h, co-immunoprecipitation assay was performed. **i** GST-pull down assay examining the interaction between GST-PPDPF, GST-PPDPF mut with Raptor. **j** The ubiquitination of Raptor in 293T cells was examined using immunoprecipitation with Raptor antibody. All experiments were repeated 3 times independently.

NAFLD. However, several medical therapies for NAFLD targeting various disease pathways have been developed, including Elafibranor (a dual PPARα/δ agonist), 6-ethylchenodeoxycholic acid (OCA) (a synthetic bile acid and FXR activator), CVC (a dual antagonist of CCR2 and CCR5), SEL (GS-4997) (a selective inhibitor of ASK1)[35]. These medications mainly target NASH, aiming to improve hepatic inflammation, liver injury, and fibrosis. However, the results of phase IIb RCT were not satisfying. Since excessive lipid deposition in the liver is a main characteristic of NAFLD, the molecules significantly affecting hepatic lipid metabolism may serve as potential therapeutic candidates for NAFLD.

In the present study, we reported that PPDPF-LKO mice spontaneously developed fatty liver at 8 months, and increased

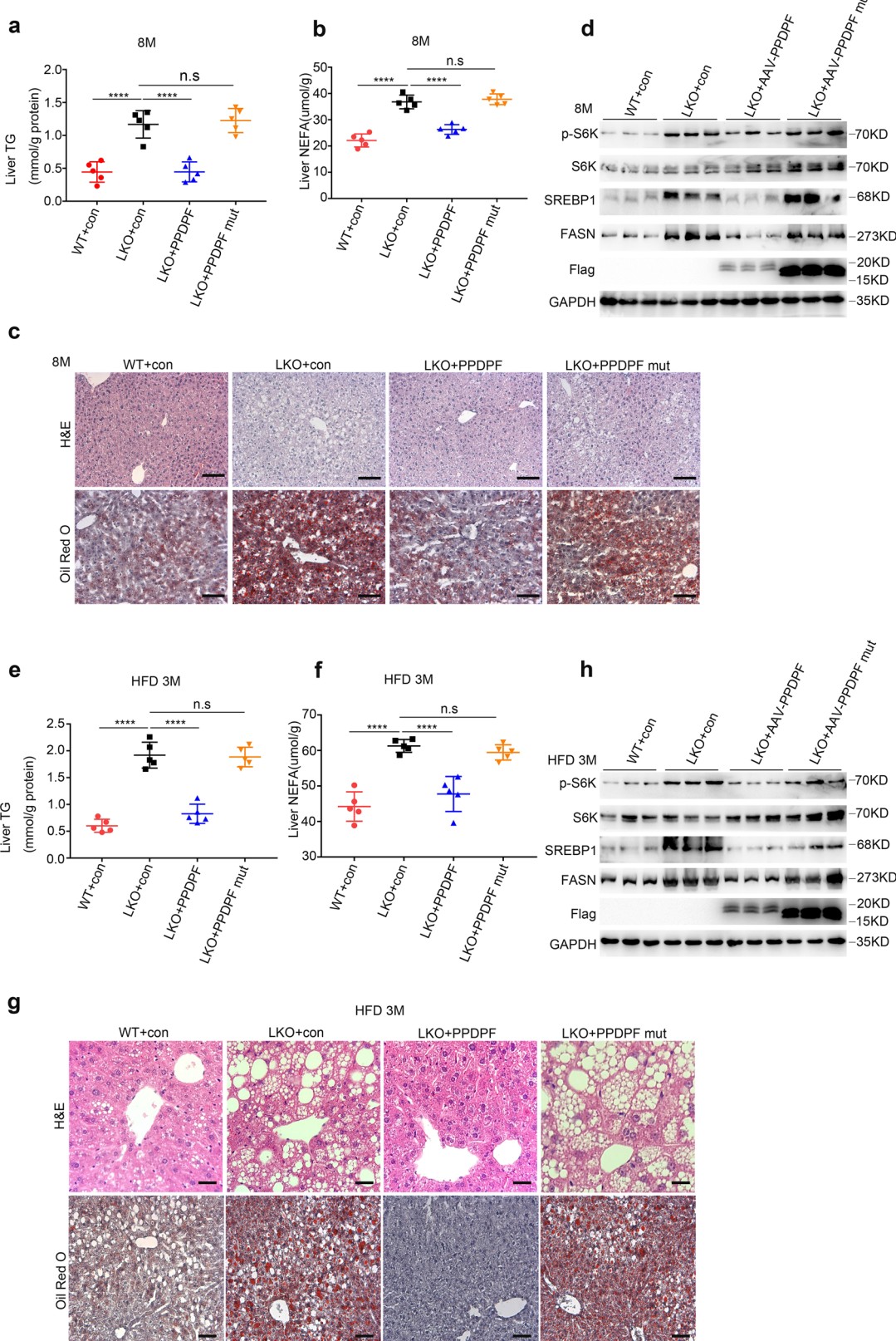

steatosis was detected in PPDPF-LKO mice compared with WT mice fed HFD. Reintroduction of PPDPF in PPDPF-LKO mice effectively prevented spontaneous fatty liver, as well as HFD-induced NAFLD. Moreover, PPDPF overexpression significantly reduced HFD-induced steatosis. These data suggested that PPDPF may be a promising drug candidate for NAFLD.

PPDPF was initially reported to regulate the development of pancreas, and this manuscript also indicated that PPDPF was upregulated in a variety of cancers[19]. Recent research found that increased expression of PPDPF in HCC tissues and patients with higher PPDPF expression had poorer prognosis[20]. Another study revealed PPDPF as the target of circular RNA circ-FOXM1 and

**Fig. 6 AAV8-mediated reintroduction of PPDPF and PPDPF-mut in PPDPF-LKO mice.** The liver TG test (**a**) and NEFA test (**b**) of WT + con, LKO + con, LKO + PPDPF, and LKO + PPDPF mut mice ($n = 5$ per group) at 8 months on chow diets. Mean ± SEM, n.s (not significant), ****$p < 0.0001$ by two-tailed unpaired Student's $t$ test. **c** Representative images of H&E and Oil Red O staining of liver sections from the mice injected with indicated adenovirus at 8 months. Scale bars, 100 um. **d** Expression of p-S6K, S6K, SREBP1 and FASN in the liver tissues of WT + con, LKO + con, LKO + PPDPF and LKO + PPDPF mut mice ($n = 3$ per group) fed HFD for 3 months. The liver TG test (**e**) and NEFA test (**f**) of WT + con, LKO + con, LKO + PPDPF, and LKO + PPDPF mut mice ($n = 5$ per group) fed HFD for 3 months. Mean ± SEM, n.s (not significant), ****$p < 0.0001$ by two-tailed unpaired Student's $t$-test. **g** Representative images of H&E and Oil Red O staining of liver sections from the mice injected with indicated adenovirus fed HFD for 3 months. Scale bars, 100 um. **h** Expression of p-S6K, S6K, SREBP1, and FASN in each group in HFD mouse model. All experiments were repeated three times independently.

facilitates cell progression in NSCLC[21]. However, the function of PPDPF remains largely unknown. Our study was the first to report the roles and molecular mechanisms of PPDPF in the development of NAFLD. Our findings suggest that PPDPF suppresses hepatic lipid synthesis by modulating mTOR signaling. PPDPF not only affects lipid deposition in the liver, but also influences HFD-induced glucose intolerance and insulin resistance. Therefore, our study revealed the physiological role of PPDPF, and expanded our understanding of this gene.

NAFLD pathogenesis is considered to be associated with mTOR signaling pathway[36]. The mTOR protein is the PI3K-like serine/threonine-protein kinase that is conserved in all eukaryotes[17,18]. mTOR is assembled into two distinct complexes, mTORC1 and mTORC2[37,38]. mTORC1 is sensitive to rapamycin, regulates cell growth and metabolism, and its activity is regulated by both nutrients and growth factors. mTORC2 is activated by growth factors and regulate cell survival, cell metabolism, and cytoskeletal organization[22,34,36,39]. It was indicated that SREBP1 activation by AKT depends on mTORC1 but not mTORC2, and accumulating evidence suggested that mTOR-S6K-SREBP1 signaling was closely involved in de novo lipid synthesis in liver[15]. mTORC1 hyperactivation by overfeeding promoted lipogenesis by inducing SREBP-1c cleavage and activation, leading to activation of lipogenic genes, such as FASN. Although the function of mTORC1 in glucose and amino acid metabolism has been largely studied, its role in lipid metabolism and its response to lipid signal remains largely unknown. In our study, we identified PPDPF as a negative regulator of mTORC1 in lipid synthesis. mTOR signaling pathway was activated in the livers of PPDPF-LKO mice, whereas it was suppressed by PPDPF overexpression. Torin1 treatment antagonized the increase of hepatic lipid accumulation in hepatocytes from PPDPF-LKO mice, and Rapamycin treatment also alleviated the symptoms of fatty liver in PPDPF-LKO mice, while had little effect on WT mice. Moreover, the expression of p-p70 S6K, SREBP1 and FASN, key lipogenic molecules downstream of mTORC1 was also suppressed by PPDPF. Taken together, these data suggested that PPDPF regulated Hepatic de novo lipogenesis through its impact on mTOR-S6K-SREBP signaling.

Raptor is a unique member of mTORC1.Previous studies have shown that ubiquitination of Raptor by DDB1-CUL4B is essential for mTORC1 activation at lysosome[23,27,40–42], and the ubiquitin hydrolase UCH-L1 can mediate Raptor deubiquitination, leading to impaired mTORC1 activity[27]. Interestingly, we found that Raptor was quickly ubiquitinated in response to PA treatment. Both basal and PA-induced ubiquitination of Raptor were significantly inhibited by PPDPF. PPDPF interacted with DDB1 and disrupted the interaction between Raptor and DDB1, which subsequently impeded DDB1-CUL4B -mediated ubiquitination of Raptor. We also detected dramatically increased Raptor-DDB1 association in PPDPF-LKO liver, indicating that enhanced Raptor-DDB1 interaction induced by PPDPF loss may contribute to hyperactivation of mTOR-S6K-SREBP signaling and development of NAFLD. In addition, we identified the 51–64 aa in PPDPF was critical for its interaction with Raptor. The PPDPF

mut with alanine substitution of these amino acids lost its ability to interact with Raptor, and therefore could not regulate mTOR signaling and lipid metabolism. This finding provides potential therapeutic target for NAFLD treatment.

In conclusion, the current study provides the first evidence that PPDPF inhibits hepatic steatosis via suppression of mTORC1 activity by interfering with Raptor–DDB1 interaction. Therefore, specifically targeting hepatic PPDPF or the PPDPF-Raptor-mTOR axis may be a promising strategies for the treatment of NAFLD.

## Methods

**Human liver samples.** The paraffin sections of fatty liver and normal liver tissues ($n = 3$) were purchased from Shanxi Avilabio Company. Our study has been approved by the Biomedical Research Ethics Committee, Shanghai Institute for Biological Sciences, CAS in 2019 (ER-SIBS-25902).

**Animals.** The PPDPF-flox mice were generated from the Model Animal Research Center of Nanjing University. A conditional PPDPF knockout mice was generated as follows. The same loxp sites were inserted on both sides of exon1-5 by homologous recombination (the second loxp was inserted via frt-neo-frt-loxp cassette). When crossed with mice carrying cre recombinase, offspring carrying both PPDPF-Flox gene and cre gene can be obtained. The expression of cre recombinase will cause exon1-5 to be deleted. The identification primers in Supplementary Table 3.

The hepatocyte-specific PPDPF knockout mice were generated as follows. The PPDPF-flox mice crossbreeding with Albumin-cre mice (Jackson Laboratory), successful PPDPF knockout and PPDPF expression could verify by qPCR.

All animal studies were approved by Institutional Animal Care and Use Committee Institutional Animal Care and Use Committee of Shanghai Institute of Nutrition and Health, CAS (IACUC, SINH, CAS), and our work was carried out in accordance with approved protocol.

**High-fat (HFD) diet-induced mice model.** HFD models were established by feeding mice on a HFD (fat, 60 kcal%; protein, 20 kcal%; carbohydrates, 20 kcal%; Research Diet) continuously for 8 weeks, 12 weeks, and 16 weeks. All animal experimental protocols were approved by Institutional Animal Care and Use Committee Institutional Animal Care and Use Committee of Shanghai Institute of Nutrition and Health, Chinese Academy of Sciences (CAS) (IACUC, SINH, CAS), and our work was carried out in accordance with approved protocol.

**Mouse adeno-associated virus8 construction and injection.** The AAV8 system was used to overexpression PPDPF in mouse livers. We amplified the PPDPF and cloned it into pAAV-TBG-3xFlag which was kindly provided by Qiurong Ding (Chinese Academy of Sciences). Then created the pAAV-TBG-PPDPF-3xFlag, pAAV-TBG-3xFlag as the control. The AAV8 virus was generated by transfecting three plasmids in 293T cells. The virus is concentrated by lodixanol (sigma, 1343517) density gradient centrifugation. Titers of the vector genome were measured by qPCR with vector-specific primers. The mice were injected with 100ul of virus-containing $2 \times 10^{11}$ AAV8 vector genomes via the tail vein. This animal experiment was approved by Institutional Animal Care and Use Committee Institutional Animal Care and Use Committee of Shanghai Institute of Nutrition and Health, Chinese Academy of Sciences (CAS), and our work was carried out in accordance with approved protocol.

**Rapamycin treatment.** Eight-week-old male mice were fed a chow diet for 6 months, injection of rapamycin (2 mg/kg body weight) was performed every other day and the mice were sacrificed at 2 months after the first injection. This animal experimental protocol was approved by Institutional Animal Care and Use Committee at Shanghai Institute of Nutrition and Health, Chinese Academy of Sciences (CAS).

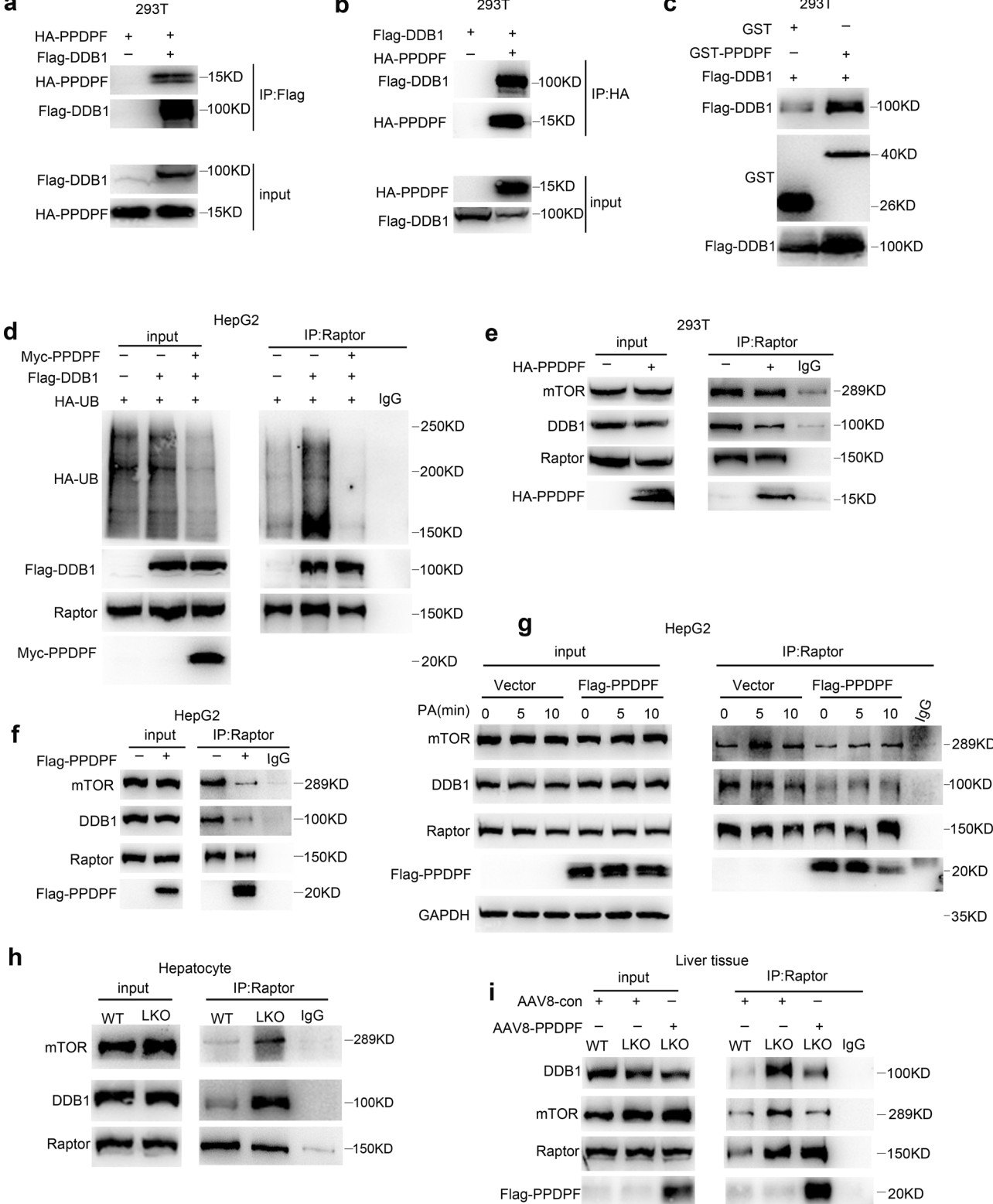

**Fig. 7 PPDPF disrupts the interaction between Raptor and DDB1. a**, **b** The interaction between PPDPF and DDB1 is detected by Co-immunoprecipitation assay in 293T cells co-transfected with Flag-DDB1 and HA-PPDPF. **c** GST pull-down is performed to confirm the interaction between PPDPF and DDB1, purified GST as the control. **d** PPDPF inhibits DDB1-mediated increase in Raptor ubiquitination. **e**, **f** PPDPF inhibites Raptor-DDB1 and Raptor-mTOR interaction in 293T and HepG2 cells by co-immunoprecipitation with Raptor antibody. **g** The interaction between Raptor with DDB1, mTOR is examined by co-immunoprecipitation upon PA stimulation in control and PPDPF-overexpressing HepG2 cells. **h** Raptor-DDB1 and Raptor-mTOR interaction in primary hepatocytes isolated from PPDPF-WT and PPDPF-LKO mice upon PA stimulation. **i** AAV-mediated reintroduction of PPDPF in the liver of LKO mice inhibits the Raptor-DDB1 and Raptor-mTOR interaction. All experiments were repeated three times independently.

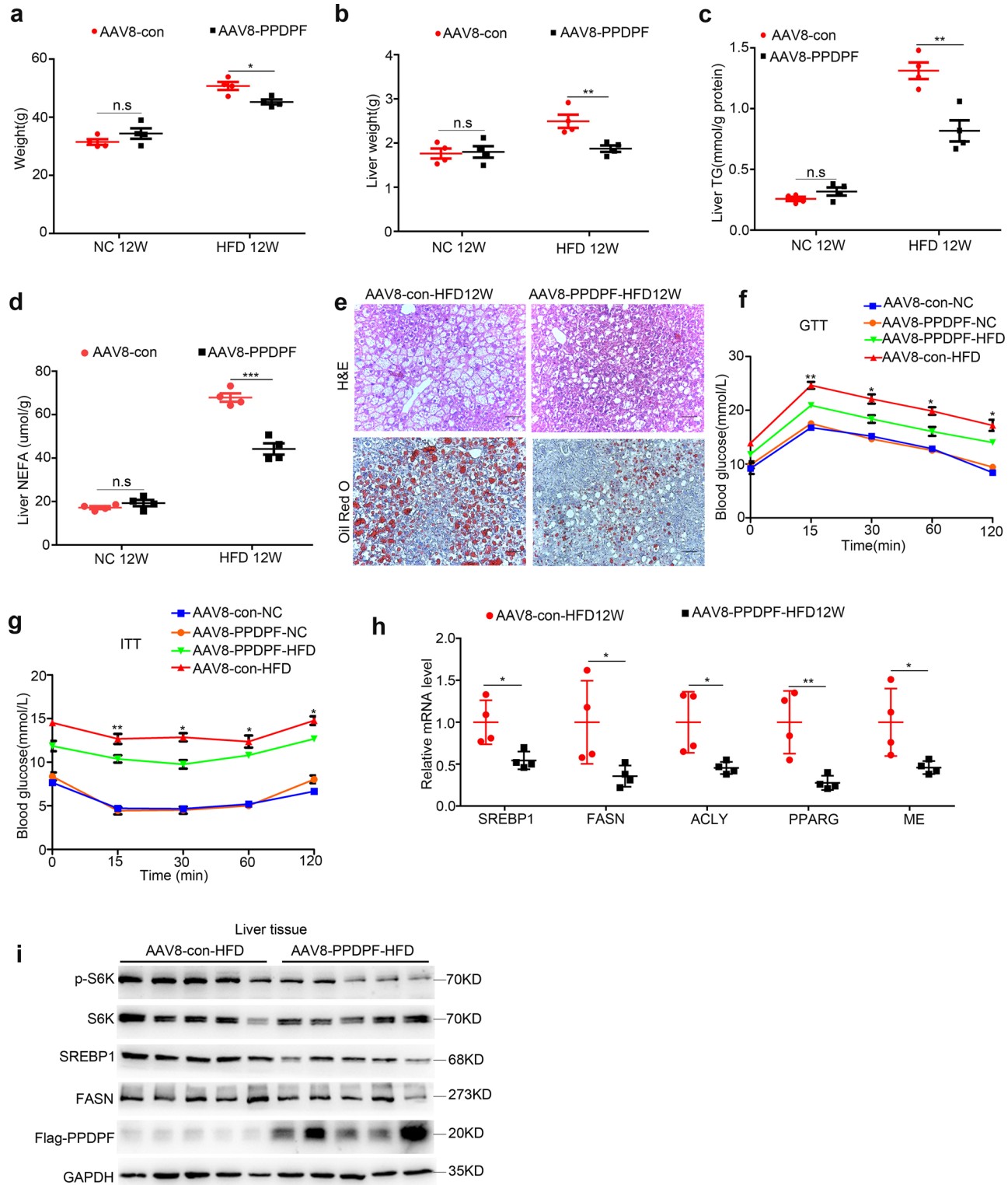

**Primary hepatocytes separation and treatment.** Primary hepatocytes were isolated from aged 8–12 weeks mice by liver perfusion. In short, mice were anesthetized with Tribromoethanol. Liver perfusion buffer was perfused through the portal vein, followed by the perfusion of liver digest buffer for 15 min. After this, the liver was excised, minced and filtered through a cell sieve (70 μm). Hepatocytes were separated via centrifugation at $1000 \times g$/min for 3 min, purified on 50% percoll. The cells were cultured in DMEM supplemented with 10% FBS and 1% penicillin–streptomycin in a 5% $CO_2$ incubator at 37 °C. Palmitate (0.4 mm; Sigma) was added to the medium for different time point.

**Cell lines.** The Human Embryonic Kidney 293T cell line was purchased from the Type Culture Collection of Chinese Academy of Sciences (Shanghai, China), and the HepG2 cell line was kindly provided by Yu Li (Chinese Academy of Sciences). All cell lines were cultured in DMEM, 10% FBS, and 1% penicillin-streptomycin, all were maintained in a humidified chamber with 5% $CO_2$ at 37 °C.

A stable monoclonal PPDPF gene overexpression HepG2 cell line was created using the flow sorting technology by lentivirus-mediated gene transfection. Briefly, we firstly cloned the PPDPF gene into lentivirus expression vector (P23-3xFlag-GFP). Next, HepG2 cell line was infected with packaged lentivirus for 8 h, followed by GFP sorting.

**Fig. 8 PPDPF overexpression reduces HFD-induced hepatic steatosis.** Body weight (**a**) and liver weight (**b**) of AAV8-con and AAV8-PPDPF mice fed a chow diet of HFD for 12 weeks (*n* = 4 for each group). Mean ± SEM, n.s (not significant), *p = 0.014 (**a**); n.s (not significant), **p = 0.01 (**b**) by two-tailed unpaired Student's *t* test. The triglyceride (TG) (**c**) and nonesterified fatty acid (NEFA) (**d**) levels in the livers of AAV8-con and AAV8-PPDPF mice at the end of 12-week HFD feeding (*n* = 4 mice for each group). Mean±SEM, n.s (not significant), **p = 0.004 (**c**); n.s (not significant), ***p = 0.0003 (**d**) by two-tailed unpaired Student's *t* test. **e** Representative H&E and Oil Red O staining of liver sections from AAV8-con and AAV8-PPDPF mice after 12-week HFD feeding. Scale bars, 100 um. GTTs (**f**) and ITTs (**g**) were performed in AAV8-con (*n* = 3) and AAV8-PPDPF (*n* = 3) mice fed a chow diet or AAV8-con (*n* = 3) and AAV8-PPDPF (*n* = 4) mice on HFD diet for 10 weeks and 11weeks. Mean ± SEM. See Supplementary Data 2 for statistics. **h** The mRNA levels of lipogenic genes in the livers of AAV8-con (*n* = 4) and AAV8-PPDPF (*n* = 4) mice on HFD diet. Mean ± SEM, *SREBP1*: *p = 0.018, *FASN*: *p = 0.045, *ACLY*: *p = 0.026, *PPARG*: **p = 0.009, *ME*: *p = 0.038 by two-tailed unpaired Student's *t* test. **i** Expression of p-S6K, S6K, SREBP1, and FASN in the liver samples of AAV8-con and AAV8-PPDPF micefed HFD for 3 months (*n* = 4 per group). All experiments were repeated three times indenpendently.

**qPCR.** qPCR was performed according to the protocol of our laboratory[43,44]. Briefly, total mRNA was isolated from cultured cells or liver samples using TRIzol reagent (Invitrogen), according to the manufacturer's instructions. 3ug cDNA was reverse transcribed into cDNA using Promega Kit (Promega, Madison) according to the manufacturer's protocol. SYBR Green (YEASEN Biotech) was applied to quantify PCR amplification. Expression levels were calculated using the Δct-method. The primer pairs used in our study are described in Supplementary Table 2.

**Plasmid constructs.** The full-length region of human PPDPF or PPDPF-mutant were cloned into the p23-Flag-GFP, pcDNA3-HA, or pcDNA3.1-Myc vector separately. The full-length region of human DDB1 was cloned into the p23-Flag-GFP vector. The full-length region of human Raptor was cloned into the p23-Flag-GFP vector. All gene fragments are obtained by PCR anplification. The primers used for plasmid construction are listed in Supplementary Table 4.

**Western blot.** Briefly, HEK293T cells and HepG2 cells were washed twice by ice-cold PBS, and lysed in lysis buffer (50 mM Tris-HCl, PH = 7.4 with 150 mM Nacl, 1 mM EDTA, and 1% Triton X-100) with protease inhibitors for 15 min on ice, after centrifugation at $20,000 \times g$ for 15 min at 4 °C. Then, protein concentration determination by Broadford. Next, protein were separated using 8% and 12% SDS-PAGE gels and then transferred to PVDF membranes (0.45 μm). After 80 min, the membranes were blocked in 5% BSA, they were incubated overnight at 4 °C with primary antibodies and then for 70 min at room temperature with the mouse and rabbit secondary antibodies. The protein levels were quantified using Tanon-5200 multifunctional chemiluminescence instrument and normalized to the levels of GAPDH. The antibodies used in our study are described in Supplementary Table 1.

**GST fusion protein purification and GST pulldown.** GST-PPDPF or GST-PPDPF mutant were purified from BL21 according to the manual. Briefly, coding sequences were subcloned into pGEX-4T-1 vector. Vectors were transformed into BL21 bacteria. Grow the bacteria in LB medium with Ampicillin to A600 0.8. IPTG was added to a final concentration of 0.3 mM, continuing incubation for additional 10 h at 30 °C. Bradford reagent was used to determine the concentration of the purified protein. The lysis of cultured cells and then pulled into purified protein, 4 °C incubated overnight. The second day, incubated with GST beads (GE) for 1 h at 4 °C and then analyzed with western blotting using the indicated antibodies.

**Mouse serum and hepatic lipid analyses.** To measure the serum TG, animals were starved for 6 h before tail vein blood collection. Blood serum was further collected after centrifugation at $8000 \times g$ for 20 min at 4 °C. For lipid tests in liver tissues, we homogenized 30 mg of liver tissue in PBS, and then adding methanol and chloroform (1:2) for extraction. Total TG were measured with a total triglyceride kit (Shanghai Shensuo). NEFA test according to the instructions of the kit (Wako, Japan).

**Nile red straining.** The Nile red was dissolved in acetone, and diluted 1:2000 in 1 × PBS for Nile Red straining. Briefly, cells on slides were fixed in 4% for-maldehyde for 15 min, followed washing three times, 5 min per time. Then staining with prepared nile red for 30 min at 37 °C, after that, washing three times, 5 min per time. Next, staining with the hocheast for 3 min, washing three times, 5 min per time. Finally, covered the film and photographed by confocal microscope (Zeiss, LSM 510 NLO).

**Histological analysis.** Firstly, put the removed liver tissue into 4% formalin for fixation. After dehydration and waxed, the liver sections were embedded in par-affin. Cut the tissues into slices (5 m), and then stained using hematoxylin and eosin (H&E) to visualize the pattern of lipid accumulation. Secondly, took part of tissues into Tissue-Tek OCT compound and became frozen liver sections. Lipid droplet accumulation in the liver was presented using Oil Red O staining.

**Statistical analyses.** All data were analyzed using GraphPad Prism 7. Quantitative values are presented as the mean ± S.E.M. Statistical differences between two experimental groups were analyzed by *t* test. A *P* value < 0.05 was considered significant.

**Reporting summary.** Further information on research design is available in the Nature Research Reporting Summary linked to this article.

## Data availability

All data supporting the findings of this study are provided within the paper and its supplementary information. All additional information will be made available upon reasonable request to the authors. Source data are provided with this paper.

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

## Acknowledgements

This work was supported by the National Key R&D Program of China (2018YFC1604404 and 2018YFC1603002), National Natural Science Foundation of China (82030084 and 81730083) to Dong Xie; and National Natural Science Foundation of China (31771538 and 81972757), Youth Innovation Promotion Association of Chinese Academy of Sciences fund (2017324), and Sanofi-SIBS 2018 Young Faculty Award to Jing-Jing Li. The authors appreciate Professor Yu Li (SINH, CAS) and Professor Jiamu Du (Southern University of Science and Technology) for their helpful suggestions on this study and Professor QiuRong Ding (SINH, CAS) for the gift of AAV8 constructs. The authors thank the New World Group for their Charitable Foundation to establish the Institute for Nutritional Sciences, SIBS, CAS-New World Joint Laboratory, which has given full support to this study.

## Author contributions

N.M., Y.W., J.L., and D.X. conceived the project and designed the research studies. N.M. and Y.W. performed most of the experiments described. S.X., Q.N., Q.Z., B.Z., H.C., H.J., and F.Z. provided help with animal and technical assistance in the mouse experiments. Y.Y., E.Z., T.C., J.X., X.D., Z.C., X.Z., and K.W. assisted with the in vitro assays. S.C., Z.L., Y.Y., X.W., and B.Z. provided conceptual advice and helpful discussion. N.M. and Y.W. analyzed data. N.M., Y.W., J.L., and D.X. wrote the manuscript.

## Competing interests

The authors declare no competing interests.
