## [Peer Review File · Nature Communications]

Reviewer comments, first round:

Reviewer #1 (Remarks to the Author):

This study reports that PPDPF inhibits hepatic steatosis via suppression of mTORC1 activity, suggesting that PPDPF may be a potential therapeutic candidate for NAFLD. Overall, the research is logical, well-designed and conclusion supported by solid data. My comments follow:

1. I would recommend to include the PPDPF expression levels in result 3 after AAV8-TBG-PPDPF virus via tail vein injection. This data is critical for determining whether the rescue effect of PPDPF has physiological or pharmacological relevance.
2. Fig. 3 (a-d) is the result after HFD consumption, which is inconsistent with the description in the main text.
3. The result in Extended Data Fig.4c shows no significant difference. It is premature to conclude that levels of genes involved in lipid treatment are decreased upon Torin1 treatment in PPDPF-KO cells.
4. All the RT-PCR results of control group need to be normalized.

Reviewer #2 (Remarks to the Author):

The authors seek to understand the molecular mechanism underlying how PPDPF alleviates hepatic steatosis in part through inhibiting Cull4/DDB1-mediated ubiquitination of Raptor to suppress mTORC1 kinase activity. The paper is clearly written, however, the following concerns should be addressed before its publication at Nature Communications.

1. Figure S1a, it will be nice for the authors to include IB against PPDPF to confirm KO efficiency.
2. Figure 1a-b, as previous studies showed that inactivation of HIPPO signaling led to increased liver growth, it will be important for the authors to monitor HIPPO signaling in WT vs PPDPF LKO.
3. Figure 1C, it will be important for the authors to examine whether PPDPF expression levels are reduced in human liver steatosis clinical samples?
4. Figure 2f, it will be nice for the authors to include IHC against PPDPF.
5. Figure 3a, it will be nice for the authors to include a Raptor interaction-deficient mutant form of PPDPF as a negative control to show the important role of activating mTOR pathway in causing fatty liver.
6. Figure 3a-f, it will be important for the authors to include the control vs PPDPF in WT as they did in the LKO conditions. This will allow the authors to examine whether reinforced expression of PPDPF can conversely, protect from HFD-induced fatty liver?
7. Figure 4e-f, it will be nice for the authors to include a Raptor interaction-deficient mutant form of PPDPF as a negative control to show the important role of PPDPF in activating mTOR through Raptor.
8. Figure 5a, it will be important for the authors to demonstrate PPDPF interaction with Raptor at endogenous levels.
9. Figure 5a-g, it is important to identify the critical residue in PPDPF that mediates the interaction with Raptor and to further show that disrupting PPDPF interaction with Raptor abolishes its ability to interfere with Raptor ubiquitination and mTORC1 signaling.
10. Figure 7, given the authors showed that loss of PPDPF activates mTOR to cause susceptibility to fatty liver, it is important for the authors to show that Rapamycin can alleviate PPDPF LKO induced fatty liver in Fig.1.

Point-by-point response to the reviewer's comments.

Reviewer's Comments:

Reviewer #1 (Remarks to the Author):

This study reports that PPDPF inhibits hepatic steatosis via suppression of mTORC1 activity, suggesting that PPDPF may be a potential therapeutic candidate for NAFLD. Overall, the research is logical, well-designed and conclusion supported by solid data. My comments follow:

1. I would recommend to include the PPDPF expression levels in result 3 after AAV8-TBG-PPDPF virus via tail vein injection. This data is critical for determining whether the rescue effect of PPDPF has physiological or pharmacological relevance.
2. Fig. 3 (a-d) is the result after HFD consumption, which is inconsistent with the description in the main text.
3. The result in Extended Data Fig.4c shows no significant difference. It is premature to conclude that levels of genes involved in lipid treatment are decreased upon Torin1 treatment in PPDPF-KO cells.
4. All the RT-PCR results of control group need to be normalized.

Reviewer #2 (Remarks to the Author):

The authors seek to understand the molecular mechanism underlying how PPDPF alleviates hepatic steatosis in part through inhibiting Cull4/DDB1-mediated ubiquitination of Raptor to suppress mTORC1 kinase activity. The paper is clearly written, however, the following concerns should be addressed before its publication at Nature Communications.

1. Figure S1a, it will be nice for the authors to include IB against PPDPF to confirm KO efficiency.

2. Figure 1a-b, as previous studies showed that inactivation of HIPPO signaling led to increased liver growth, it will be important for the authors to monitor HIPPO signaling in WT vs PPDPF LKO.
3. Figure 1C, it will be important for the authors to examine whether PPDPF expression levels are reduced in human liver steatosis clinical samples?
4. Figure 2f, it will be nice for the authors to include IHC against PPDPF.
5. Figure 3a, it will be nice for the authors to include a Raptor interaction-deficient mutant form of PPDPF as a negative control to show the important role of activating mTOR pathway in causing fatty liver.
6. Figure 3a-f, it will be important for the authors to include the control vs PPDPF in WT as they did in the LKO conditions. This will allow the authors to examine whether reinforced expression of PPDPF can conversely, protect from HFD-induced fatty liver?
7. Figure 4e-f, it will be nice for the authors to include a Raptor interaction-deficient mutant form of PPDPF as a negative control to show the important role of PPDPF in activating mTOR through Raptor.
8. Figure 5a, it will be important for the authors to demonstrate PPDPF interaction with Raptor at endogenous levels.
9. Figure 5a-g, it is important to identify the critical residue in PPDPF that mediates the interaction with Raptor and to further show that disrupting PPDPF interaction with Raptor abolishes its ability to interfere with Raptor ubiquitination and mTORC1 signaling.
10. Figure 7, given the authors showed that loss of PPDPF activates mTOR to cause susceptibility to fatty liver, it is important for the authors to show that Rapamycin can alleviate PPDPF LKO induced fatty liver in Fig.1.

Point-by-point response to the reviewer's comments.

Reviewer #1 (Remarks to the Author):

This study reports that PPDPF inhibits hepatic steatosis via suppression of mTORC1 activity, suggesting that PPDPF may be a potential therapeutic candidate for NAFLD. Overall, the research is logical, well-designed and conclusion supported by solid data.

My comments follow:

1. I would recommend to include the PPDPF expression levels in result 3 after AAV8-TBG-PPDPF virus via tail vein injection. This data is critical for determining whether the rescue effect of PPDPF has physiological or pharmacological relevance.

Response:

The PPDPF expression levels in result 3 was shown in Fig.4e and 4f.

Figure4

2. Fig. 3 (a-d) is the result after HFD consumption, which is inconsistent with the description in the main text.

Response:

We apologize for the inaccurate description in the figure legends of Figure 3 and Extended Data Figure 3. The Fig.3 (a-f) is the result of the 8-month-old mice fed a chow diet. The Extended Data Figure 3 is the result after HFD consumption. We have corrected the description in the Figure legends of Figure 3 and Extended Data Figure 3.

Figure 3

Fig.3 AAV8-mediated reintroduction of PPDF rescues the phenotype of PPDF-null in PPDF-LKO mice fed a chow diet for 8 months.

(a-d) The body weight(a), liver weight(b), TG(c) and NEFA(d) for each group (n=4

per group) at 8 months on chow diets.

(e) Representative images of H&E and Oil Red O staining in liver sections of mice injected with indicated adenovirus at 8 months. Scale bars, 100 μ m.

(f) The mRNA expression level of lipogenesis-related genes (n=6 per group).

Extended Data Figure 3

Extended Data Fig.3 AAV8-mediated reintroduction of PPDPF rescues

PPDPF-null phenotype in PPDPF-LKO mice fed HFD for 3 months.

(a-d) The body weight(a), liver weight(b), TG(c) and NEFA(d) for each group (n=6 per group) at 3 months after HFD consumption.

(e) Representative images of H&E and Oil Red O staining in liver sections of mice injected with indicated adenovirus at 8 months. Scale bars, 100 um.

(f) The mRNA expression level of lipogenesis-related genes (n=4 per group).

3. The result in Extended Data Fig.4c shows no significant difference. It is premature to conclude that levels of genes involved in lipid treatment are decreased upon Torin1 treatment in PPDPF-KO cells.

Response:

We treated the cells with Torin1 for 2 hours before, which may contribute to the subtle difference. However, the change of the expression of the genes for lipid synthesis was only detected in PPDPF-LKO cells upon Torin1 treatment, while their expression was not influenced by Torin1 in WT cells at 2h. Furthermore, we performed Torin1 treatment for 18 hours, and the expression of the genes for lipid synthesis was significantly decreased in both WT and PPDPF LKO cells, and the alteration was more dramatic in PPDPF-LKO primary hepatocytes. These data suggest that PPDPF LKO cells are more sensitive to Torin1, which was consistent with increased activation of mTOR signaling in PPDPF LKO cells. The results are shown below:

Extended Data Figure 4

C

4. All the RT-PCR results of control group need to be normalized.

Response:

As suggested, all the RT-PCR results of control group have been normalized.

Reviewer #2 (Remarks to the Author):

The authors seek to understand the molecular mechanism underlying how PPDPF alleviates hepatic steatosis in part through inhibiting Cull4/DDB1-mediated ubiquitination of Raptor to suppress mTORC1 kinase activity. The paper is clearly written, however, the following concerns should be addressed before its publication at Nature Communications.

1. Figure S1a, it will be nice for the authors to include IB against PPDPF to confirm KO efficiency.

Response:

As suggested, we examined the knockout efficiency of PPDPF in mouse liver tissue by qPCR and western blotting (Extended Data Figure 1a).

Extended Data Figure 1

2. Figure 1a-b, as previous studies showed that inactivation of HIPPO signaling led to increased liver growth, it will be important for the authors to monitor HIPPO signaling in WT vs PPDPF LKO.

Response:

Thanks for the reviewer's suggestion. Activation of the Hippo pathway leads to phosphorylation of YAP (p-YAP), resulting inhibition of YAP. Thus we detected the level of p-YAP in the livers of 8-month-old mice by western blotting. As shown below, knockout of PPDPF promoted the phosphorylation level of YAP, which indicated that the HIPPO signaling pathway was inhibited. This was in contrast to the

increased liver weight of PPDPF-LKO mice. We mainly investigate the role of PPDPF in liver lipid metabolism in this study, so we do not include this result in our manuscript.

3. Figure 1C, it will be important for the authors to examine whether PDDPF expression levels are reduced in human liver steatosis clinical samples?

Response:

Following this suggestion, we detected the expression of PPDPF in the liver tissues of normal and NAFLD patients by immunohistochemistry. The results (Figure1h) show that expression of PPDPF is downregulated in the livers of NAFLD patients.

Figure 1

4. Figure 2f, it will be nice for the authors to include IHC against PPDPF.

Response:

Thanks for the reviewer's suggestion. However, the anti-mouse PPDPF antibody for immunohistochemistry is not commercially available. Instead, we examined the knockout efficiency of PPDPF by western blots (Extended Data Figure2a):

Extended Data Figure 2

a

5. Figure 3a, it will be nice for the authors to include a Raptor interaction-deficient mutant form of PPDPF as a negative control to show the important role of activating mTOR pathway in causing fatty liver.

Response:

According to this suggestion, we have found a Raptor interaction-deficient mutant form of PPDPF by screening of multiple PPDPF mutants. In Fig.3, we have not yet demonstrated the signal pathways affected by PPDPF, so these results were included in Fig.6 and Extended Data Figure 7. These data demonstrate that the Raptor interaction-deficient mutant of PPDPF loses its ability to regulate mTOR signaling, lipid metabolism and development of NAFLD.

Figure6

Fig.6 AAV8-mediated reintroduction of PPDPF and PPDPF-mut in PPDPF-LKO mice.

(a, b) The liver TG(a) and NEFA(b) for each group (n=5 per group) at 8 months on chow diets.

(c) Representative images of H&E and Oil Red O staining in liver sections of mice injected with indicated adenovirus at 8 months. Scale bars, 100 μ m.

(d) Expression of p-S6K, S6K, SREBP1 and FASN.

(e, f) The liver TG(e) and NEFA(f) for each group (n=5 per group) at HFD diet for 3 months.

(g) Representative images of H&E and Oil Red O staining in liver sections of mice injected with indicated adenovirus at HFD diet for 3 months. Scale bars, 100 μ m.

(d) Expression of p-S6K, S6K, SREBP1 and FASN at HFD diet for 3 months.

Extended Data Figure 7

Extended Data Fig.7 AAV8-mediated reintroduction of PPDF and PPDF-mut

in PPDPF-LKO mice fed a chow diet for 8 months.

(a, b) The body weight(a), liver weight(b), for each group (n=5 per group) at 8 months on chow diets.

(c)The mRNA expression level of lipogenesis-related genes (n=5 per group).

(d, e) The body weight(d), liver weight(e), for each group (n=5 per group) at 3 months after HFD diets.

(f)The mRNA expression level of lipogenesis-related genes at 3 months after HFD diets. (n=5 per group).

6. Figure 3a-f, it will be important for the authors to include the control vs PPDPF in WT as they did in the LKO conditions. This will allow the authors to examine whether reinforced expression of PPDPF can conversely, protect from HFD-induced fatty liver?

Response:

Thanks for this suggestion. In Figure3, PPDPF was overexpressed in the LKO condition to confirm its role in the development of NAFLD. The protective effect of PPDPF against NALFD was demonstrated in Figure8, which included the experiments suggested by the reviewer.

Figure 8

Fig.8 PPDPF overexpression reduces HFD-induced hepatic steatosis

(a and b) Body weight (a) and liver weight (b) of AAV8-CON and AAV8-PPDPF mice fed for 12 weeks with an NC and HFD (n=4 mice for each group).

(c and d) The triglyceride (TG) (c) and nonesterified fatty acid (NEFA) (d) levels in the livers of AAV8-CON and AAV8-PPDPF mice at the end of 12 weeks of HFD feeding (n=4 mice for each group).

(e) Representative H&E and Oil Red O staining of liver sections from AAV8-CON and AAV8-PPDPF mice after 12 weeks of HFD feeding. Scale bars, 100 μ m.

(f and g) GTTs (f) and ITTs (g) were performed on AAV8-CON and AAV8-PPDPF mice after NC or HFD feeding for 10 weeks and 11 weeks respectively (n=4 in each group).

(h) The mRNA levels of lipogenesis genes in the livers of the indicated groups (n=4 in each group). All experiments were repeated 3 times; data represent the mean \pm SEM. * P<0.05, ** P<0.01, *** P<0.001.

(i) Expression of p-S6K, S6K, SREBP1 and FASN in liver samples after HFD induced 3 months (n=4 per group).

7. Figure 4e-f, it will be nice for the authors to include a Raptor interaction-deficient mutant form of PPDPF as a negative control to show the important role of PPDPF in activating mTOR through Raptor.

Response:

We have performed the experiments suggested by the reviewer, and the results are shown in Fig.6d and Fig.6h.

Figure 6

8. Figure 5a, it will be important for the authors to demonstrate PPDPF interaction with Raptor at endogenous levels.

Response:

As suggested, we examined the interaction between PPDPF and Raptor at endogenous level in primary hepatocytes, and the result is shown in Fig.5c and Fig5d.

Figure 5

9. Figure 5a-g, it is important to identify the critical residue in PPDPF that mediates the interaction with Raptor and to further show that disrupting PPDPF interaction with Raptor abolishes its ability to interfere with Raptor ubiquitination and mTORC1 signaling.

Response:

As suggested, we identified 51-64aa was the critical residue in PPDPF that mediated the interaction with Raptor. The 51-64aa (GHWASFFFGKSTL) was mutated to (AAAAAAAAAAAAAAAA) (PPDPF mut). Subsequently, we transfected WT and PPDPF mut constructs into 293T and HepG2 cells, respectively. We found that disrupting PPDPF interaction with Raptor indeed abolished its ability to interfere with Raptor ubiquitination and mTORC1 signaling. The results are shown in Fig.5 and Extended Data Fig.6.

Fig.5 PPDPF interacts with Raptor and inhibits the ubiquitination of Raptor

(a and b) A Co-immunoprecipitation assay performed in 293T cells co-transfected with Flag-Raptor and HA-PPDPF to detect the interaction.

(c and d) Western blot results showing endogenous PPDPF interacted with Raptor.

(e) PPDPF interacts with Raptor in vitro. Purified GST was used as a control.

(f) GST-pull down assay showing the 51-64aa interacted with Raptor.

(g and h) 293T cells were transfected with control vector and PPDPFWT, PPDPF mut, HA-Raptor plasmid, respectively. After 48h, immunoprecipitation assay was

performed.

(i) GST-pull down assay showing the interaction between GST-PPDPF, GST-PPDPF mut with Raptor.

(j) The ubiquitination of Raptor in 293T cells was examined using immunoprecipitation with Raptor antibody.

Extended Data Figure 6

Extended Data Fig.6 Influence of PPDPF and PPDPF mut on Raptor ubiquitination and mTOR signaling pathway.

(a) The ubiquitination of Raptor in HepG2 cells was examined using immunoprecipitation with Raptor antibody.

(b) The dynamic change of Raptor ubiquitination in HepG2 cells under PA treatment.

(c) The ubiquitination of Raptor in WT and LKO hepatocytes was examined using

immunoprecipitation with Raptor antibody.

(d) Expression of p-S6K, S6K, SREBP1 and FASN in HepG2 cells after PA induced.

10. Figure 7, given the authors showed that loss of PPDPF activates mTOR to cause susceptibility to fatty liver, it is important for the authors to show that Rapamycin can alleviate PPDPF LKO induced fatty liver in Fig.1.

Response:

Following this advice, we treated both WT and PPDPF LKO mice with Rapamycin for 2 months, and the results are shown in Extended Data Figure 5. As expected, Rapamycin treatment significantly decreased the weight, liver weight, liver TG content, liver NEFA level, lipid droplets and the expression of the genes involved in lipid synthesis in PPDPF LKO mice, while it showed little effect on WT mice.

Extended Data Figure 5

Extended Data Fig.5 Rapamycin treatment inhibites lipid synthesis in

PPDPF-LKO mice at 8 months of age on chow diets.

(a-d) The body weight(a), liver weight(b), TG(c) and NEFA(d) for each group (n=4 per group) at 8 months on chow diets.

(e) Representative images of H&E and Oil Red O staining in liver sections of mice injected with rapamycin treatment at 8 months. Scale bars, 100 um.

(f)The mRNA expression level of lipogenesis-related genes (n=4 per group).

(g)Expression of p-S6K, S6K, SREBP1 and FASN in liver samples after rapamycin treatment.

Reviewer comments, second round:

REVIEWERS' COMMENTS

Reviewer #2 (Remarks to the Author):

The authors have addressed most of the raised concerns during this round of revision.

Point-by-point response to the reviewer's comments.

Reviewer #2 (Remarks to the Author):

The authors have addressed most of the raised concerns during this round of revision.

Response: We appreciate the reviewer's help and effort in reviewing our manuscript.